# Randomized nutrient bar supplementation improves exercise-associated changes in plasma metabolome in adolescents and adult family members at cardiometabolic risk

Michele Mietus-Snyder[1]*, Nisha Narayanan[2], Ronald M. Krauss[3,4], Kirsten Laine-Graves[5], Joyce C. McCann[5], Mark K. Shigenaga[5], Tara H. McHugh[6], Bruce N. Ames[5], Jung H. Suh[5]

1 Division of Cardiology, Department of Pediatrics, Children's National Hospital, George Washington University School of Medicine and Health Sciences, Washington DC, United States of America, 2 Weill Cornell Medical College, Cornell University, New York, New York, United States of America, 3 University of California Benioff Children's Hospital San Francisco, San Francisco, California, United States of America, 4 Children's Hospital Oakland Research Institute, Oakland, California, United States of America, 5 University of California Benioff Children's Hospital Oakland Research Institute, Oakland, California, United States of America, 6 United States Department of Agriculture, Western Regional Research Center, Albany, California, United States of America

☯ These authors contributed equally to this work.
* mmsnyder@childrensnational.org

## Abstract

### Background

Poor diets contribute to metabolic complications of obesity, insulin resistance and dyslipidemia. Metabolomic biomarkers may serve as early nutrition-sensitive health indicators. This family-based lifestyle change program compared metabolic outcomes in an intervention group (INT) that consumed 2 nutrient bars daily for 2-months and a control group (CONT).

### Methods

Overweight, predominantly minority and female adolescent (Teen)/parent adult caretaker (PAC) family units were recruited from a pediatric obesity clinic. CONT (8 Teen, 8 PAC) and INT (10 Teen, 10 PAC) groups randomized to nutrient bar supplementation attended weekly classes that included group nutrition counseling and supervised exercise. Pre-post physical and behavioral parameters, fasting traditional biomarkers, plasma sphingolipids and amino acid metabolites were measured.

### Results

In the full cohort, a baseline sphingolipid ceramide principal component composite score correlated with adiponectin, triglycerides, triglyceride-rich very low density lipoproteins, and atherogenic small low density lipoprotein (LDL) sublasses. Inverse associations were seen between a sphingomyelin composite score and C-reactive protein, a dihydroceramide composite score and diastolic blood pressure, and the final principal component that included glutathionone with

**Data Availability Statement:** All relevant data are within the manuscript and its Supporting Information files.

**Funding:** The authors thank Rachelle Woods and the USDA Agricultural Research Services Western Regional Research Center (Albany, CA) for assisting with nutrient bar production, the staff at the University of California Benioff Children's Hospital Oakland Pediatric Clinical Research Center for professional assistance with study visit assesssments, and Teresa Klask for invaluable administrative assistance. The authors are grateful to several organizations for their generous donations of bar ingredients: Pharmachem Laboratories (Kearny, NJ, USA) for a vitamin and mineral blend; the U.S. Highbush Blueberry Council (Folsom, CA, USA) for blueberry powder; and Dow Chemical (Midland MI, USA) for hydroxypropylmethylcellulose. This work was supported by the Children's Hospital Oakland Research Institute-Ames Foundation (M.M-S., J.C. M., M.K.S.), Quest Diagnostics (R.M.K.), and Mass Spectrometry Core supported by NIH S10OD0018070 (J.H.S.). The funders had no role in study design, data collection and analysis, decision to publish, or preparation of the manuscript.

**Competing interests:** The authors have declared that no competing interests exist.

fasting insulin and the homeostatic model of insulin resistance. In CONT, plasma ceramides, sphinganine, sphingosine and amino acid metabolites increased, presumably due to increased physical activity. Nutrient bar supplementation (INT) blunted this rise and significantly decreased ureagenic, aromatic and gluconeogenic amino acid metabolites. Metabolomic changes were positively correlated with improvements in clinical biomarkers of dyslipidemia.

## Conclusion

Nutrient bar supplementation with increased physical activity in obese Teens and PAC elicits favorable metabolomic changes that correlate with improved dyslipidemia.

The trial from which the analyses reported upon herein was part of a series of nutrient bar clinical trials registered at clinicaltrials.gov as NCT02239198.

## Introduction

Cardiovascular disease (CVD) remains the leading cause of mortality in the United States [1], and after decades of decline, is rising coincident with the increase in obesity, insulin resistance, and diabetes that characterize cardiometabolic risk (CMR) [2]. Notwithstanding hereditary predisposition, reduction in identified, modifiable lifestyle risk factors can reverse CMR and CVD [3]. It is estimated that 45.4% of all cardiometabolic deaths in the United States due to heart disease, stroke, and diabetes are associated with suboptimal intakes (too much or too little) of 10 dietary factors [4].

Fewer than 1% of American children and adolescents meet full recommended metrics of heart healthy nutrition, falling especially short of recommended intake in the categories of fruits, vegetables, fiber and essential fatty acids [5, 6]. Intensive pediatric lifestyle interventions for obesity are effective in achieving significant reduction in body mass index (BMI) but do not elicit stable changes in nutrition habits in children and adolescents [7]. These studies suggest a critical need for developing innovative tools to improve diet quality in youth.

We have previously shown that twice daily consumption for two weeks of a whole food based nutrient bar composed of a blueberry, dark chocolate, red grape, and walnut matrix, soluble and insoluble fiber, with supplemental vitamins, minerals and essential long chain fatty acids, significantly increased high density lipoprotein cholesterol (HDL), due primarily to a 28% increase in large HDL particles, in generally healthy and insulin sensitive lean and overweight adults [8]. In a subsequent 2 mo study of the effects of the nutrient bar on CMR markers in individuals across a range of BMIs, only those with low inflammation at baseline as assessed by high sensitivity C-reactive protein (CRP) < 14.3 nmol/L (<1.5 mg/L) responded comparably to those in the earlier trial, with not only increased HDL cholesterol and large HDL particles but also a trend toward increased high molecular weight adiponectin and a decrease in other CMR factors at 2 weeks, sustained at 2 months (9). In particular, a shift in low density lipoprotein (LDL) particle subfractions toward a less atherogenic profile was evident in the noninflamed group (decreased small and medium LDL and increased large LDL). Although the participants with overweight or obesity and CRP > 14.3 nmol/L did not show this response, they did experience an upward trend in adiponectin by 2 months [9]. These results suggest that there may be a continuum of metabolic responsiveness to this nutrient supplement that is slowed in the face of the chronic low-level inflammation commonly observed with obesity and insulin resistance. A 6 month study of this nutrient bar in obese adolescents with non-eosinophilic asthma showed improved lung function at 2 months, but favorable movement in cardiometabolic biomarkers only began to emerge at 6 months [10].

It is not known whether more sensitive biomarkers of early metabolic change may be capable of detecting short-term effects of the nutrient bar supplementation in persons at CMR. Past metabolomics studies in obese adolescents and adults have identified strong positive associations between baseline levels of branched chain, aromatic, sulfur, and gluconeogenic amino acid metabolites and parameters of inflammation and insulin resistance [11, 12]. Similarly, elevated levels of specific ceramide species have been shown to associate with inflammation, dyslipidemia and insulin resistance in both adult and adolescent obesity [13]. The relative sensitivity of these biomarkers to reflect moderate changes in dietary intake of polyphenols, essential lipids, fiber and vitamin/minerals, remains incompletely understood.

In the present randomized, controlled, non-blinded trial, a two month intervention with exercise and nutrition counseling alone (CONT) or with nutrient bar supplementation (INT) was performed in a high CMR cohort of adolescent (Teen)/parent adult caretaker (PAC) family units to determine 1) cross-sectional relationships in both adolescents and adults between traditional CMR biomarkers and amino acid and ceramide metabolites and 2) longitudinal changes within groups in the same CMR biomarkers following the lifestyle +/- nutrient bar intervention.

## Methods

### Study population

This study was approved by the University of California (UC) Benioff Children's Hospital Oakland Human Subjects Review Board, under approval number 2011022 entitled: The Impact of a Nutritional Supplement on Weight and Metabolic Health in a Parent-Child Intervention (S1 Protocol). All participating adolescents signed an assent document and all parent /adult legal guardian participants signed a written consent for their own participation and a separate consent for their participating adolescent(s). The study cohort was recruited from the UC San Francisco Benioff Children's Hospital Oakland weight management program (Fig 1). Teen participants were eligible for inclusion if they had a BMI greater than the 95th percentile, were between 14 and 18 years of age, fluent in English (as group lifestyle counseling was conducted in English), and willing to eat the nutrient bar twice daily, after having tasted a sample, with one or both PACs also willing to participate. PAC inclusion criteria were the same except that there was no adult weight threshold. Exclusion criteria for both Teens and PACs included the diagnosis of diabetes mellitus (type 1 or 2) or hypertension (>140/90 for adults, >95th percentile + 5 mm Hg for Teens), and use of glucocorticoid, weight loss, insulin-sensitizing, lipid-lowering, or anti-hypertensive medication.

With Institutional Review Board approval, PAC consent for self and adolescent, and adolescent assent, 17 dyads and 1 triad (19 PAC, 18 Teens) were enrolled (Fig 1 and S1 Flowchart) in May and June, 2011. One INT PAC dropped out before study measures were initiated but the rest of the family unit continued participation. All family units were seen within two weeks of the baseline group counseling session and within 2 weeks of the final session on the Clinical and Translational Research unit for pre-post blood testing. Participants were randomized at the baseline visit 1.25:1 to INT:CONT groups that met on separate days in the late afternoon after school and work for eight identical lifestyle counseling sessions in July and August 2011; 30 minutes of weekly group nutrition counseling followed by 30 minutes of supervised group exercise. In addition, the INT group was given a one week supply of nutrient bars at each visit through the seventh week and advised to return wrappers at the following session. Bar composition has been previously described (Bar2, in Table 1, reference [9]), and was designed to supplement prevalent nutrient deficiencies in the American diet up to daily recommended intake (RDI), including 515 mg polyphenols, 4 gm whey protein, 9 gm of total fiber (soluble and

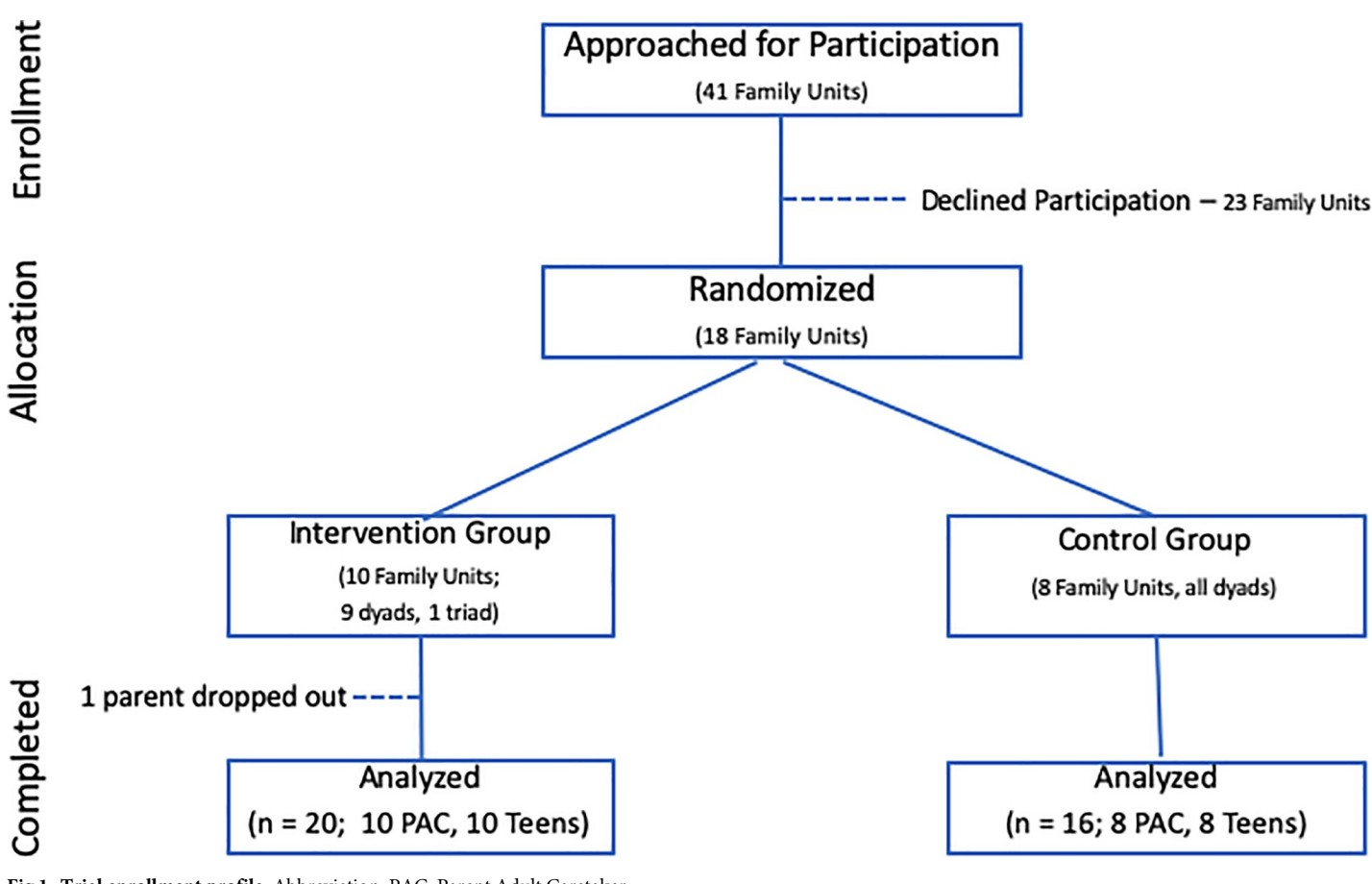

**Fig 1. Trial enrollment profile.** Abbreviation: PAC, Parent Adult Caretaker.

insoluble) and 200 mg of long chain omega 3 fatty acid docosahexaenoic acid (DHA). Most vitamins and minerals added to the whole food matrix of the bar are in amounts representing 10% to 50% of their corresponding recommended daily intake, with the exceptions of vitamin C (105 mg) and vitamin D (2300 IU) both added above RDI. Additional vitamin C serves as an antioxidant preservative for other bar ingredients, notably the omega 3 fatty acid DHA. Additional vitamin D addresses the prevalent deficiency in persons at high CMR, and the relatively low blood levels in the study cohort (Table 1).

Consumption of 2 bars each day was advised, with the first to be eaten before noon and the second in either the afternoon or evening, together with a minimum of 8 ounces of water with each bar due to the high fiber content. Compliance with eating the supplement (INT group) and home lifestyle adherence (both INT and CONT groups) were evaluated with phone call or text communication to each participating family every other day. Pre-post change in 25 hydroxy vitamin D levels served as an objective measure of compliance with the nutrient bars.

## Procedures and assays

Assessment of physical (weight, height, waist circumference, blood pressure, resting heart rate), behavioral (self-report diet by Block Food Frequency Questionnaire (Nutritionquest, Berkeley CA) [14] and activity (8 questions of usual weekly activity) [15], and metabolic (CMR biomarkers listed below) status was conducted at baseline and study completion on all Teen and PAC participants.

**Table 1. Baseline participant profiles.**

| | CONTROL (n = 16; 81% Female) 19% HW, 19%HAI, 6% HB, 6%NHW, 50% NHB | | | INTERVENTION (n = 20; 75% Female) 40% HW, 40% HAI, 10% HB, 5% NHW, 5% NHB | | |
|---|---|---|---|---|---|---|
| | PAC (N = 8) | Teen (N = 8) | CONT Combined (N = 16) | PAC (N = 10) | Teen (N = 10) | INT Combined (N = 20) |
| Age (yrs) | 41.9 (12.2) | 15.3* (1.3) | **28.6 (16.1)** | 43.0 (3.8) | 15.6* (1.4) | **29.3 (14,3)** |
| Weight ** (Kg) | 103.7 (14.9) | 102.9 (27.9) | **103.3 (21.7)** | 83.0 (18.6) | 85.3 (8.7) | **84.1 (14.2)** |
| Height (m) | 1.64 (0.08) | 1.69 (0.06) | **1.66 (0.07)** | 1.61 (0.07) | 1.63 (0.09) | **1.62 (0.08)** |
| Waist Circ. (cm) | 116.0 (9.1) | 106.9 (19.3) | **111.4 (15.3)** | 102.0 (16.9) | 103.0 (10.3) | **102.5 (13.7)** |
| BMI ** (Kg/m$^2$) | 38.6 (5.2) | 35.9 (8.1) | **37.2 (6.7)** | 31.8 (5.9) | 32.2 (4.6) | **32.0 (5.2)** |
| WHRatio | 70.7 (5.8) | 63.3 (10.6) | **67.0 (9.1)** | 63.2 (9.7) | 63.4 (9.1) | **63.4 (9.1)** |
| TC (mmol/L) | 5.1 (1.1) | 4.2 (0.8) | **4.7 (1.0)** | 5.2 (0.5) | 4.5 (1.1) | **4.9 (0.9)** |
| TG (mmol/L) | 1.3 (0.6) | 1.4 (0.9) | **1.3 (0.7)** | 2.0 (1.8) | 1.2 (0.4) | **1.6 (1.4)** |
| HDL (mmol/L) | 1.3 (0.3) | 1.1 (0.2) | **1.2 (0.2)** | 1.4 (0.3) | 1.2 (0.3) | **1.3 (0.3)** |
| LDL (mmol/L) | 3.2 (1.0) | 2.5 (0.6) | **2.8 (0.8)** | 3.0 (0.4) | 2.8 (0.9) | **2.9 (0.7)** |
| TGHDLR | 2.4 (1.7) | 2.9 (2.1) | **2.7 (1.9)** | 4.3 (6.3) | 2.5 (1.1) | **3.4 (4.5)** |
| Vit D (nmol/L) | 37.4 (10.0) | 49.7 (21.2) | **44.0 (16.9)** | 48.0 (9.0) | 53.4 (16.2) | **50.7 (15.2)** |
| Glucose (mmol/L) | 5.5 (0.2) | 5.4 (0.3) | **5.4 (0.3)** | 5.4 (0.4) | 5.1 (0.3) | **5.3 (0.4)** |
| Insulin (pmol/L) | 126.0 (81.) | 179.3 (127.3) | **154.8 (107.9)** | 76.8 (27.8) | 154.1 * (77.0) | **115.8 (69.1)** |
| HOMA | 5.25 (3.6) | 7.2 (5.5) | **6.3 (4.5)** | 3.1 (1.3) | 5.9 * (3.0) | **4.5 (2.7)** |
| SBP (mm Hg) | 127 (15) | 112 (8) * | **120 (14)** | 120 (11) | 117 (8) | **119 (9)** |
| DBP (mm Hg) | 77 (9) | 66 (7)* | **72 (10)** | 79 (10) | 65 (9)* | **72 (12)** |
| HR | 67 (13) | 75 (7) | **71 (11)** | 72 (9) | 70 (10) | **71 (9)** |
| CRP ** (nmol/L) | 43.8 (25.7) | 57.1 (30.5) | **50.5 (27.6)** | 31.4 (16.2) | 28.6 (22.8) | **30.5 (20.0)** |
| Adipo (ng/ml) | 1556 (1389) | 1313 (588) | **1427 (1007)** | 2507 (1654) | 1296 (587) | **1933 (1381)** |

Values are mean (standard deviation).

* Denotes statistically significant ($p$ <0.05) difference by unpaired student's t-test between PAC (Parent Adult Caretakers) and Teen subgroups within Control (CONT) and/or Intervention (INT) groups.

** Denotes statistically significant ($p$ <0.05) differences for the designated variable by unpaired student's t-test between combined CONT and INT groups
Abbreviations: n–number; HW–Hispanic White; HAI–Hispanic American Indian; HB–Hispanic Black; NHW–Nonhipanic White; NHB–Nonhispanic Black; yrs–years; Kg- kilograms; m–meters; mmol/L–millimoles per liter; nmol/L–nanomoles per liter; mm Hg–millimeters of mercury; ng/ml–nanograms per milliliter; Waist Circ–Waist Circumference; TGHDLR–Triglyceride to HDL ratio; SBP–Systolic blood pressure, DBP–Diastolic blood pressure, TC- Total Cholesterol; TG–Triglyceride; LDL-Low Density Lipoprotein cholesterol; HDL–High Density Lipoprotein cholesterol; Vit D– 25 hydroxy vitamin D; HOMA–Homeostatic model assessment of insulin resistance; CRP–high sensitive C-reactive protein; HR–resting heart rate; Adipo- High Molecular Weight Adiponectin.

Anthropometric and clinical evaluation: All anthropometric measures were performed in duplicate in the clinical research center and if not within 10% agreement, were repeated a third time. The reported measure is an average of the two closest numbers. Height was measured with a stationary stadiometer. Weight was measured using a digital electronic scale and the BMI and waist to height ratio (WHRatio) were calculated. Waist circumference (Waist Circ) was measured at end expiration to the nearest mm with a Gulick II Plus tape midway between the lowest border of the rib cage and the upper border of the iliac crest.

Blood pressure and Resting Heart Rate: Each was measured in triplicate after 5 minutes sitting quietly with readings taken at least one minute apart. An automatic digital blood pressure monitor was used with cuff size adjusted for arm size.

Traditional CMR biomarkers: Fasting blood samples were drawn and processed in the UCSF Benioff Children's Hospital Oakland Clinical Research Center and sent to ARUP Diagnostic Laboratories for: standard lipid profile [total triglyceride, total cholesterol, and cholesterol within HDL and LDL (calculated by the Friedwald formula with reflex to direct LDL), glucose, insulin, 25 hydroxy Vitamin D level and CRP. TG to HDL ratio and non-HDL were calculated. Fasting insulin and glucose were used to calculate the Homeostasis Model Assessment of Insulin Resistance Index (HOMA) according to the formula: fasting insulin (microU/L) x fasting glucose (nmol/L)/22.5. Lipoprotein particle subclasses were analyzed by an ion mobility procedure that sensitively and directly measures concentrations of lipoprotein particle subfractions [16]. High molecular weight adiponectin was measured by solid-phase sandwich ELISA (Quantikine; R&D Systems, Minneapolis, MN).

Metabolomic analyses: 1) Targeted analyses of 42 amine-containing metabolites consisting of 20 major amino acids, and secondary metabolites of arginine and cysteine whose levels are sensitive to inflammation and oxidative stress were performed on stored samples preserved at -70° Fahrenheit. Briefly, plasma was acidified with 5% perchloric acid containing 8 stable isotope internal standards. Acid-soluble supernatant was used for strong-cation exchange solid-phase extraction to capture cationic amine- containing metabolites. Extracted metabolites were further derivatized with isopropylchloroformate. Derivatives of metabolites were resolved using Agilent 1260 ultra-high pressure liquid chromatography and eluted with a gradient of water and isopropanol (65% v/v). An Agilent 6490 triple quadrupole mass spectrometer was used to detect resolved analytes and quantify them using authenticated external and internal standards. 2) Sphingolipidomics by electrospray tandem mass-spectrometry by validated techniques [17] was used to identify sphingolipid metabolites, including ceramides.

## Statistical analysis

De-identified baseline and study completion data points, paired by participant study ID, were entered into SPSS (IBM Statistics, Seattle, WA). Descriptive analyses of the study cohort were summarized and results for Teens and PACs, as well as for combined CONT and INT groups, were compared by unpaired Student t-test. Continuous physical and metabolic variables were tested for normality by examining the skewness, kurtosis and the Shapiro Wilk tests and transformed as necessary before analysis. Most of the variables in our data were normally distributed. Log transformations were conducted for the continuous physical and metabolic variables that were skewed to make them as normal as possible. The Shapiro-Wilk tests show (with $p$-value $> 0.05$) that all of the transformed variables except two (i.e. small and medium LDL particle concentrations) are approximately normal. The measures of metabolite concentrations used for principal component analysis (PCA) were Z-transformed to render them normally distributed on the same scale with mean of zero and standard deviation of one. Pre-post change in absolute metabolite levels were compared by repeated samples paired t-test. Baseline z-scores of metabolites were subjected to principal component analysis (PCA) without rotation. PCA is an unsupervised analysis that aims to decrease the complexity of data by reducing variables to a smaller number of principal components (PC). Each of the PCs were vectors of metabolite contributions. A direct oblimin rotation was used and 6 factors before the bend in the scree plot (S1 Fig), and eigenvalues $>1$ were retained. Component scores for each participant were calculated with a standardized scoring coefficient. A PCA model with oblique rotation was tested to examine the factor correlation matrix. Since none of the factor correlations

were over ± 0.32, indicating factor correlations are not driven by the data, we estimated a model with orthogonal rotation to reveal a simplified structure with interpretable factor loadings.

To identify metabolite patterns of interest, bivariate Pearson-correlation analysis was performed between percent changes in metabolites and clinical biomarkers of CMR (dyslipidemia and insulin resistance) and with lipoprotein particle subfraction distributions separately. A generalized estimating equation procedure determined the significance of longitudinal changes in PC scores and their contributing metabolites using age and gender as co-variates. Wald Chi-square tests determined the significance of pairwise differences in treatment responses. Statistical analyses were performed using IBM SPSS Statistics version 25 (IBM, Armonk, NY, USA) and Rstudio version 1.1.456, and $p$-values <0.05 were considered statistically significant for the differences in traditional biomarkers. A Bonferroni correction for multiple comparisons adjusted the $p$-value set as significant to <0.002 for the correlational analyses with amino acid and ceramide metabolites.

## Results

### Cohort description

The study cohort was predominantly female, 61% Hispanic (half who identify as White and half as American Indian), 25% Nonhispanic Black, 8% Hispanic Black. and 6% Nonhispanic White, with more Hispanic Not Black than Nonhispanic Black participants randomized by chance to the INT group (Table 1). Full assessment of physical, behavioral, metabolic, and metabolomic status was conducted at baseline and study completion on the cohort of 36 participants (S1 Data).

There was excellent attendance in both INT and CONT groups with all Teens and PAC participating in more than 78% of group sessions and 100% of baseline and follow-up assessment visits. There was considerable prevalence of obesity, dyslipidemia, inflammation and insulin resistance in all participants and mild hypertension in adult participants at baseline (Table 1). Mean baseline BMI was over 30 mg/m² in all participants, did not differ by sex, but was significantly higher in the CONT group relative to INT. The mean CRP was over 28.6 nmol/L (3 mg/L) in all participants but higher in Teen participants assigned to the CONT group than to the Teen INT group, and between the combined CONT and INT groups. Despite these random differences in BMI and CRP, central adiposity (WHtRatio), blood pressure, lipids, insulin resistance and inflammation were similar in both groups, suggesting closely matched CMR profiles. Although in population studies, Waist Circ is greater in males as compared with females, only one of the 18 PAC was male; while his Waist Circ fell below the adult female PAC mean, it was not the lowest recorded among PAC. The average Waist Circ for Teen males (106.3 ± 19.2 cm) vs female (103.8 ± 11.8 cm) Teens did not differ significantly ($p$ = 0.72) so Waist Circ data are not reported by sex. Compliance with nutrient bar intake by self-report was 85.8 ± 11.1% and 86.7 ± 13.8% among INT group adults and Teens respectively.

### Clinical changes in CVD risk factors following intervention

The quality of self-report diets assessed by Block food frequency questionnaire was poor in all participants at baseline (S2 Data). Modest but significant improvement in CONT PAC was evident in categories of saturated fat intake and added sugar (less sugared beverage) but trends in the same direction did not reach significance in INT PAC or in Teens from either group. Average daily servings of dietary fruit (1.4 for PAC, 1.1 for Teens), vegetables, not including potatoes (1.5 for PAC, 0.9 for Teens), and fiber (10 grams for PAC, 8.5 grams for Teens) were

**Table 2. Effects of interventions on anthropometric and clinical measures.**

| | CONTROL | | | | INTERVENTION | | | |
|---|---|---|---|---|---|---|---|---|
| | PAC | | Teen | | PAC | | Teen | |
| | Baseline | Post | Baseline | Post | Baseline | Post | Baseline | Post |
| Activity Score | 8.6 (6.9) | 14.8 * (2.8) | 12.1 (4.9) | 14.3 * (4.0) | 8.2 (3.6) | 10.6 * (3.5) | 10.3 (4.2) | 14.3* (3.7) |
| Vit. D (nmol/L)** | 37.4 (10.0) | 40.6 (8.5) | 49.7 (21.2) | 57.2 (18.4) | 47.9 (14.2) | 74.4 (26.7)* | 53.4 (16.2) | 72.6 (24.7)* |
| Weight (Kg) | 103.7 (14.9) | 103.6 (15.4) | 102.9 (27.9) | 103.9 (28.6) | 82.9 (18.6) | 83.2 (18.7) | 85.3 (8.7) | 86.1 (8.8) |
| Waist Circ (cm) | 115.9 (9.1) | 110.5 * (10.7) | 106.9 (19.3) | 106.6 (18.3) | 102.0 (16.9) | 101.5 (13.7) | 103.0 (10.3) | 103.1 (9.60 |
| WHRatio | 0.71 (0.06) | 0.67* (0.07) | 0.63 (0.10) | 0.63 (0.08) | 0.63 (0.11) | 0.63 (0.10) | 0.64 (0.09) | 0.63 (0.08) |
| BMI (kg/m$^2$) | 38.6 (5.2) | 38.5 (5.1) | 35.9 (8.1) | 36.1 (8.5) | 31.8 (5.9) | 31.9 (6.0) | 32.2 (4.6) | 32.5 (4.8) |
| Adipo (ng/ml) | 1556 (1389) | 1933 (1738) | 1313 (588) | 1212 (433) | 2506 (1654) | 3134 (2584) | 1296 (587) | 1193 (451) |
| CRP (nmol/L) | 43.8 (25.7) | 59.1 (51.4) | 57.1 (30.5) | 34.3 (25.7) | 31.4 (16.2) | 41.0 (29.5) | 27.6 (22.8) | 34.3 (23.8) |
| Glucose (nmol/L) | 5.5 (0.2) | 5.6 (0.5) | 5.3 (0.3) | 5.4 (0.5) | 5.4 (0.4) | 5.4 (0.4) | 5.1 (0.3) | 5.2 (0.5) |
| Insulin (pmol/L) | 21.1 (13.5) | 18.4 (8.8) | 29.9 (21.2) | 29.4 (15.9) | 12.8 (4.6) | 12.5 (7.4) | 25.8 (12.8) | 29.5 (21.9) |
| HOMA | 5.3 (3.6) | 4.6 (2.1) | 7.2 (5.5) | 7.2 (4.1) | 3.1 (1.3) | 3.1 (2.0) | 5.9 (3.0) | 6.8 (5.2) |
| TC (mmol/L) | 5.1 (1.1) | 5.0 (1.) | 4.2 (0.8) | 4.1 (0.9) | 5.2 (0.5) | 5.4 (0.6) | 4.5 (1.0) | 4.6 (1.0) |
| TG (mmol/L) | 1.3 (0.6) | 1.3 (0.6) | 1.4 (0.9) | 1.5 (1.1) | 2.0 (1.8) | 1.7 (0.9) | 1.2 (0.4) | 1.3 (0.6. |
| LDL (mmol/L) | 3.2 (1.0) | 3.1 (1.3) | 2.5 (0.6) | 2.3 (0.6) | 3.0 (0.4) | 3.1 (0.4) | 2.8 (0.9) | 2.7 (0.7) |
| HDL (mmol/L) | 1.3 (0.3) | 1.3 (0.3) | 1.1 (0.2) | 1.1 (0.2) | 1.4 (0.3) | 1.4 (0.2) | 1.2 (0.3) | 1.2 (0.)3 |
| TGHDLR | 2.4 (1.9) | 2.5 (1.7) | 2.9 (2.1) | 3.5 (2.9) | 4.3 (6.3) | 2.9 (2.2) | 2.5 (1.1) | 2.6 (1.1) |
| SBP **T (mm Hg) | 127 (15) | 124 (15) | 112 (8) | 117 (9) | 120 (11) | 119 (14) | 117 (8) | 109* (10) |
| DBP (mm Hg) | 77 (9) | 80 (9) | 66 (7) | 65 (7) | 79 (10) | 76 (8) | 65 (9) | 65 (8) |
| HR | 67 (13) | 69 (10) | 75 (7) | 80 (8) | 72 (9) | 64 (8) | 70 (10) | 73 (11) |

Values are mean ± SD (standard deviation).

* Denotes statistically significant differences ($p < 0.05$) by paired student's t-test within a group.

** Denotes statistically significant ($p < 0.05$) time by group interactions for the designated variable for both PAC and Teens (Vitamin D levels) unless specified (T for Teens only, SBP).

Abbreviations as listed with Table 1.

comparably low in both CONT and INT groups and did not change in any age subgroup (full nutritional data are in S1 and S2 Data). Self-report activity increased in all participants in both groups (Table 2). Weight was stable, even in the INT group despite the addition of 220 kcal in two daily nutrient bars. Good compliance with nutrient bar intake was suggested by a significant increase in plasma 25 hydroxy Vitamin D (Vit D) in the INT group PAC and Teens, but neither CONT subgroup. There were no other consistent effects of nutrient bar supplementation evident in traditional CMR biomarkers, nor were there significant within- or between-group changes in any anthropometric measures in combined CONT and INT groups.

Subgroup analysis of PAC and Teens separately showed some favorable changes in both CONT and INT groups. PAC CONT participants decreased Waist Circ (Table 2). Among Teen participants, CRP trended lower in CONT but did not reach significance ($p = 0.1$). Among Teens, systolic blood pressure decreased significantly by 7% ($p = 0.03$) in INT only, trending upward in CONT Teens ($p = 0.08$) with a significant between-group difference.

## Baseline correlations between metabolomic profiles and clinical biomarkers

Previous bar supplementation trials showed that chronic inflammation (CRP > 14.3 nmol/L or 1.5) was associated with slower response in metabolic improvement. All participants, both Teens and PACs in this study, met criteria for obesity and had baseline CRP > 14.3 nmol/L.

To test the hypothesis that subtle shifts in metabolism may precede changes in traditional bio-markers, a targeted analysis of plasma ceramides and amino acids was performed.

Baseline metabolite data were z-transformed and subjected to PCA analysis. The top six PCs extracted, each composed of distinct sets of linked metabolites, explained 59.3% of the total variance in the data set (Table 3). PC1, which explained 16.7% of the variance, was composed exclusively of ceramides and ceramide-1-phophates; PC2 of amino acid metabolites; PC3 of sphingomyelins; PC4 of dihydroceramide species, glucosylceramide, and one specific ceramide subspecies Cer 18:1; PC5 of biomarkers of meat consumption (3-methylhistidine, 1-methylhistidine, and beta-alanine [18]) and transsulfuration and polyamine metabolites (cystathionine, putrescine); and PC6 of amino acids in antioxidant defense (notably glutathione).

Fig 2A shows bivariate correlational analysis between baseline PC1-PC6 scores with parameters of obesity, dyslipidemia, insulin resistance and inflammation. Within the full cohort, no significant correlations were observed between any PC scores and anthropometric measures of weight, BMI, WC, or waist-height ratio. In contrast, highly significant associations between specific PC scores and metabolic parameters of dyslipidemia, insulin resistance and inflammation were present. PC1 (ceramide-related metabolites) was positively correlated with fasting total cholesterol, triglycerides, and HDL (r = 0.41, 0.38, and 0.35 respectively, all at $p \leq 0.04$), and adiponectin (r = 0.52, p = 0.002). PC 3 (sphingomyelin metabolites) was inversely associated with CRP (r = -0.52, $p$ = 0.002). PC4 (dihydroceramide compounds) was inversely associated with diastolic blood pressure (DBP, r = -0.48, $p$ = 0.004) and PC6 (including antioxidant glutathione) showed an inverse association with fasting insulin (r = -0.34 $p$ = 0.04) and HOMA (r = -0.35, $p < 0.05$).

In Fig 2B positive association patterns are illustrated between PC1 scores and atherogenic lipoprotein particles, specifically very small LDLIVc (r = 0.53 p = 0.001), LDLIVa (r = 0.44 p = 0.009) and all three subclasses of VLDL (large, r = 0.51 p = 0.002; medium, r = 0.48 p = 0.004, and small, r = 0.45 p = 0.006). The PC1 score was also associated with very small LDL IVb (r = 0.42, $p$ = 0.11) and inversely with LDL diameter (r = -0.35, p = 0.04), both above $p$ value threshold set at 0.01. It is also of note that the baseline relationship between PC1 Cers

**Table 3. Pattern matrix of baseline principal component variables.**

| Metabolites | PC1 | Metabolites | PC2 | Metabolites | PC3 | Metabolites | PC4 | Metabolites | PC5 | Metabolites | PC6 |
|---|---|---|---|---|---|---|---|---|---|---|---|
| | (16.7) | | (15.3) | | (10.4) | | (8.5) | | (4.9) | | (3.5) |
| Cer C22:0 | 0.88 | Met | 0.76 | T. SPM | 0.98 | DHCer C22:0 | 0.89 | Cysth | 0.96 | Sar | 0.90 |
| Cer C24:1 | 0.86 | Tyr | 0.65 | SPM C22:0 | 0.97 | DHCer C26:0 | 0.87 | 3-MH | 0.95 | Asp | 0.83 |
| Cer C20:0 | 0.84 | Val | 0.61 | SPM C14:0 | 0.94 | DHCer C24:0 | 0.86 | Put | 0.90 | GSH | 0.62 |
| Cer C16:0 | 0.83 | Trp | 0.57 | SPM C16:0 | 0.91 | DHCer C16:0 | 0.80 | 1-MH | 0.74 | | |
| Cer C22:1 | 0.82 | | | SPM C24:1 | 0.87 | Cer C18:1 | 0.79 | Beta-Ala | 0.63 | | |
| T. Cer | 0.82 | | | SPM C22:1 | 0.86 | DHCer C20:0 | 0.76 | | | | |
| Cer C14:0 | 0.80 | | | SPM C24:0 | 0.85 | DHCer C14:0 | 0.76 | | | | |
| C1P C18:0 | 0.79 | | | SPM C18:1 | 0.78 | DHCer C18:0 | 0.75 | | | | |
| Cer C24:0 | 0.72 | | | SPM C20:0 | 0.68 | | | | | | |
| Cer C26:1 | 0.59 | | | | | | | | | | |

Numbers in () denote % of total variance explained by each PCs.

Numbers below PC denotes correlation between metabolites and PC noted on each column heading.

Abbreviations: PC: Principal Component, Cer: Ceramide, T. Cer: Total Ceramide, C1P: Ceramide-1-phosphate, T. SPM: Total Sphinogomyelin, SPM: Sphinogomyelin, DHCer: Dihydroceramide, GlcCer: Glucosylceramide, Met: methionine, Tyr: tyrosine, Val: valine, Trp: tryptophan, Cysth: cystathionine, 3-MH: 3-methylhistidine, Put: putrescine, 1-MH: 1-methylhistidine, Beta-Ala: beta-alanine, Sar: sarcosine, Asp: aspartate, and GSH: glutathione.

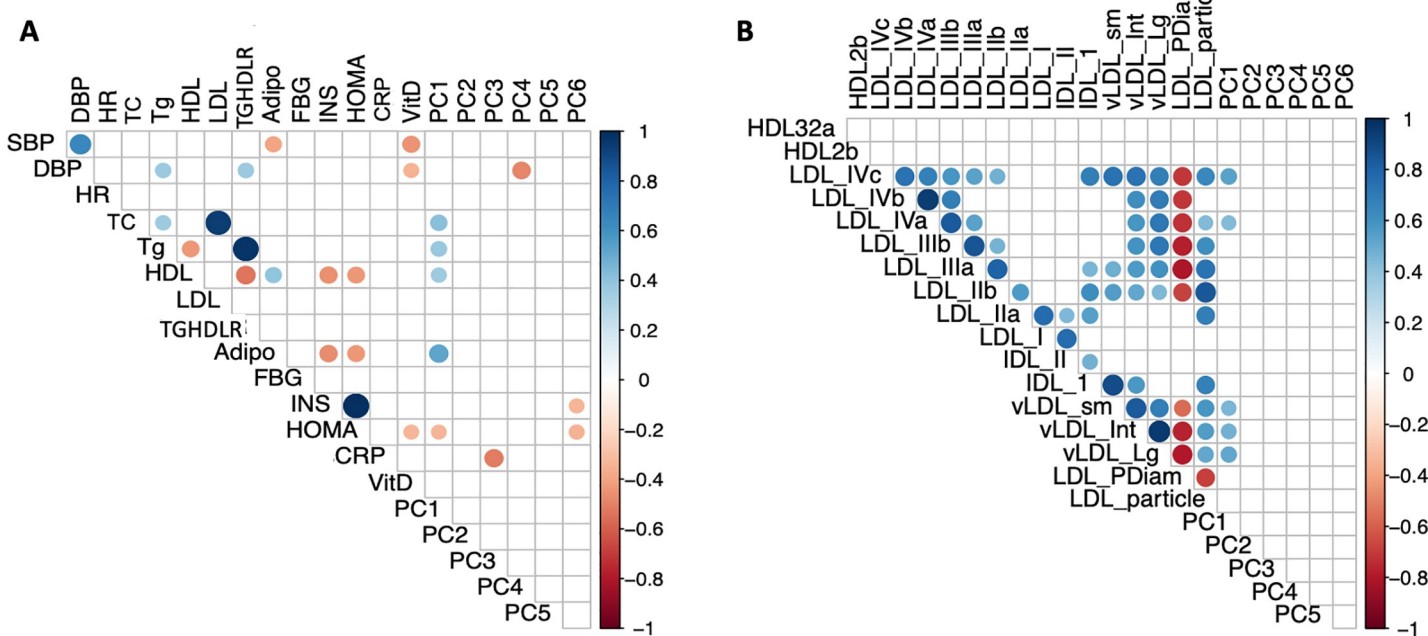

**Fig 2.** Correlograms for Principal Component (PC) scores with A. traditional biomarkers of cardiometabolic risk and B. Lipoprotein subspecies. Color-coded r values indicated on the y axes to the right. Positive correlations are depicted in blue and negative correlations in red. The strength of correlation is represented by the size as well as the intensity of the color of the spheres plotted. Distinct associations are shown between PC1 (sphingolipid ceramide) metabolites with parameters of dyslipidemia (total cholesterol, triglyceride and HDL) in 2A (shown if $p < 0.05$) and small LDL IVc and IVa particles in 2B (shown if $p < 0.01$). PC3 and PC4 showed non-overlapping associations with CRP and diastolic blood pressure (DBP), respectively. Abbreviations: SBP–Systolic blood pressure, DBP–Diastolic blood pressure, TC- Total Cholesterol; TG–Triglyceride; LDL-Low Density Lipoprotein cholesterol; HDL–High Density Lipoprotein cholesterol; TGHDLR–Triglyceride to HDL-cholesterol ratio, Adipo- High Molecular Weight Adiponectin, FGB–Fasting blood glucose, ISN–Insulin, HOMA-IR–Homeostatic model assessment of insulin resistance; CRP–high sensitive C-reactive protein; VitD– 25 hydroxy vitamin D; PC–principal component. Lipoprotein sub-species include HDL3-2a (small) and HDL2b (large); LDLIVc, IVb, IVa, IIIb (Very Small); LDLIIIa (Small); LDL IIb (Medium); and LDLIIa and I (Large); intermediate density lipoprotein–IDL, IDLII (small) and IDLI (large); very low density lipoprotein–vLDL, vLDLsm, Int, Lg (small, Intermediate and Large); Diam—Diameter.

and HDL was driven primarily by association with the small HDL32a subclass (r = 0.33, $p < 0.05$) not large HDL2b.

## Metabolomic and lipidomic changes observed with intervention

We next determined how increased physical activity by itself or with nutrient bar supplementation modulated these baseline metabolomic parameters. Analysis of PC change was performed with adjustments of potential covariates including age, gender, and baseline CRP. While there was no change in the plasma total ceramide pool, results showed significant divergence in several key ceramide species within PC1 between CONT and INT.

Results show significant increases in C14:0 (15%), C20:0 (20%), C22:0 (14%) and C24:1 (17%) ceramides and a nearly significant increase in C16:0 (14%, $p = 0.055$) in CONT. In contrast, C24:1 ceramide significantly decreased in the INT group by 18%; all other ceramides trended lower or remained unchanged (Table 4). Analysis of group by time interactions showed significant between group differences in all of these C14:0, C16:0, C20:0, C22:0 and C24:1 ceramide species.

Sphinganine and sphingosine are essential substrates for ceramide synthesis through de novo synthesis and salvage pathways. In CONT only, levels of sphinganine and sphingosine increased significantly by 59% and 50%, respectively resulting in significant pairwise differences between CONT and INT groups (Table 5).

**Table 4. Intervention effects on fasting plasma ceramides.**

| Measure | | | Mean | SD | p values | |
|---|---|---|---|---|---|---|
| | | | | | Within group | Time x group |
| Cer C14:0 | CONT | Pre | 0.06 | 0.020 | **0.050** | **0.007** |
| | | Post | 0.069 | 0.020 | | |
| | INT | Pre | 0.069 | 0.022 | 0.070 | |
| | | Post | 0.062 | 0.018 | | |
| Cer C16:0 | CONT | Pre | 0.115 | 0.036 | **0.055** | **0.024** |
| | | Post | 0.131 | 0.036 | | |
| | INT | Pre | 0.122 | 0.031 | 0.233 | |
| | | Post | 0.115 | 0.031 | | |
| Cer C18:0 | CONT | Pre | 0.212 | 0.072 | 0.727 | 0.192 |
| | | Post | 0.216 | 0.068 | | |
| | INT | Pre | 0.217 | 0.072 | 0.179 | |
| | | Post | 0.195 | 0.067 | | |
| Cer C18:1 | CONT | Pre | 0.033 | 0.008 | 0.211 | **0.058** |
| | | Post | 0.035 | 0.008 | | |
| | INT | Pre | 0.033 | 0.009 | 0.152 | |
| | | Post | 0.031 | 0.009 | | |
| Cer C20:0 | CONT | Pre | 1.094 | 0.412 | **0.019** | **0.045** |
| | | Post | 1.318 | 0.480 | | |
| | INT | Pre | 1.184 | 0.398 | 0.745 | |
| | | Post | 1.158 | 0.461 | | |
| Cer C20:1 | CONT | Pre | 0.008 | 0.004 | 0.279 | 0.325 |
| | | Post | 0.009 | 0.004 | | |
| | INT | Pre | 0.009 | 0.004 | 0.745 | |
| | | Post | 0.009 | 0.004 | | |
| Cer C22:0 | CONT | Pre | 1.635 | 0.688 | 0.104 | **0.040** |
| | | Post | 1.864 | 0.688 | | |
| | INT | Pre | 1.748 | 0.662 | 0.206 | |
| | | Post | 1.593 | 0.666 | | |
| Cer C22:1 | CONT | Pre | 0.019 | 0.008 | 0.501 | 0.117 |
| | | Post | 0.02 | 0.008 | | |
| | INT | Pre | 0.021 | 0.009 | 0.083 | |
| | | Post | 0.019 | 0.009 | | |
| Cer C24:0 | CONT | Pre | 7.317 | 2.876 | 0.534 | 0.356 |
| | | Post | 7.945 | 3.112 | | |
| | INT | Pre | 6.918 | 2.773 | 0.476 | |
| | | Post | 6.585 | 3.005 | | |
| Cer C24:1 | CONT | Pre | 0.814 | 0.372 | 0.139 | **0.014** |
| | | Post | 0.956 | 0.372 | | |
| | INT | Pre | 1.00 | 0.358 | **0.042** | |
| | | Post | 0.82 | 0.358 | | |

(*Continued*)

**Table 4.** (Continued)

| Measure | | | Mean | SD | p values | |
|---|---|---|---|---|---|---|
| | | | | | Within group | Time x group |
| T. Cer | CONT | Pre | 11.496 | 4.292 | 0.289 | 0.144 |
| | | Post | 12.781 | 4.524 | | |
| | INT | Pre | 11.534 | 4.141 | 0.288 | |
| | | Post | 10.775 | 4.365 | | |

Pre-post changes for key ceramide (Cer) species are shown for CONT (Control) and INT (Intervention) groups with p values for both within group and between group (Time x group) differences. Metabolites with significant between group differences are shaded in light gray.

CI—confidence interval of within group changes; SD–standard deviation; T. Cer- Total Ceramide pool.

Sphingosine-1-phosphate (S1P) is a terminal breakdown product of ceramide and an important anti-inflammatory and vascular signaling lipid mediator associated with lipoproteins, particularly HDL, in plasma. As shown in Table 5, S1P levels increased differentially in both CONT (31%; $p = 0.05$) and more so INT (51%; $p = 0.03$) participants. The S1P increase in the INT group differed from the degree of change in CONT at $p = 0.007$.

Table 6 lists amino acid metabolites with significant within group and between group changes. Results show that among CONT, fasting levels of proline, ornithine, lysine, alanine and threonine trended upward. In contrast, the nutrient bar INT significantly altered amino acid metabolism, such that concentrations of several gluconeogenic, sulfur redox and urea cycle intermediates decreased, notably fasting levels of serine, proline, aspartate, cystathionine, alanine, glutamine methionine, and citrulline. A small but significant increase in histidine concentration associated with favorable nitrogen balance, improved bioavailability of arginine, and an increase in the Fischer ratio the molar ratio of BCAA (leucine, valine, isoleucine) and aromatic amino acids (AAA, phenylalanine and tyrosine), indicative of favorable vascular and liver functions respectively, were observed in the INT group only.

To establish the clinical significance of differences in metabolism detected among INT participants, bivariate correlations between percent changes in ceramide and amino acid metabolites and clinical biomarkers of dyslipidemia and insulin resistance were considered (Fig 3). Positive correlations were observed between changes in TG and changes in ceramides composed of C14:0 (r = 0.6; $p = 0.005$), C16:0 (r = 0.7; $p = 0.001$), C24:1 (r = 0.7; $p = 0.0004$) and C26:1 (r = 0.6; $p = 0.007$) acyl chain lengths (Fig 3A). Specific associations were evident between percent changes in C18:1 ceramide and changes in very small LDL IVc (r = 0.61; $p = 0.004$) and IVb (r = 0.60; $p = 0.005$) particles (Fig 3B). In addition, changes in fasting blood

**Table 5. Intervention effects on plasma sphingolipid bases.**

| Sphingoid Bases | CONT | | | INT | | | Time x group p values |
|---|---|---|---|---|---|---|---|
| | Baseline (nmol/L) | Pairwise Diff (nmol/L) | 95% CI (nmol/L) | Baseline (nmol/L) | Pairwise Diff (nmol/L) | 95% CI (nmol/L) | |
| Sphingosine | 38.2 ± 4.7 | **+19.4***  | +1.0 - +37.8 | 241.9 ± 82.7 | -10.2 | -95.1 - +74.7 | 0.02 |
| Dihydro-sphingosine-1-phosphate | 5.2 ± 0.7 | +2.4 | 0.0 - + 4.7 | 4.2 ± 0.4 | +1.3 | -0.3 - + 3.0 | 0.05 |
| Sphingosine -1-phosphate | 282.9 ± 25.1 | **+88.8***  | +0.4 - +177.1 | 256.8 ± 22.7 | **+130.5***  | +38.4 - +222.5 | 0.007 |

*denotes significant within group changes (bolded) before and after intervention at $p \leq 0.002$.

Time x group denotes significance of between group changes.

CI- confidence interval of within group changes.

**Table 6. Differential changes in plasma amino acid metabolites following CONT and INT interventions.**

| Amino Acids | Control | | | INT | | | |
|---|---|---|---|---|---|---|---|
| | Baseline (μmol/L) | Pairwise Diff (μmol/L) | 95% CI (μmol/L) | Baseline (μmol/L) | Pairwise Diff (μmol/L) | 95% CI (μmol/L) | Time x Group |
| Serine | 127.9 ± 9.0 | +7.4 | -13.5 - +28.4 | 157.0 ± 10.0 | -33.4* | -50.5 - -16.3 | 0.0001 |
| Proline | 186.5 ± 9.3 | +23.9* | +2.1 - +45.8 | 249.8 ± 12.7 | -50.6* | -73.0 - -28.3 | 0.0001 |
| Aspartate | 25.1 ± 2.1 | +2.2 | -3.1 - +7.4 | 38.6 ± 2.5 | -10.9* | -16.4 - -5.3 | 0.0001 |
| Cystathionine | 1.0 ± 0.2 | -0.04 | -0.2 - +0.1 | 0.8 ± 0.04 | -0.3* | -0.4 - -0.2 | 0.0001 |
| Sarcosine | 19.2 ± 2.1 | -1.6 | -7.0 - +3.8 | 29.6 ± 1.7 | -10.1 | -14.1 - -6.2 | 0.0001 |
| Ornithine | 84.8 ± 8.3 | +41.8* | +9.9 - +73.6 | 122.1 ± 8.0 | +8.3 | -35.9 - +52.7 | 0.001 |
| Arg Bioavail Ratio | 0.6 ± 0.06 | -0.2 | -0.3 - -0.05 | 0.4 ± 0.03 | +0.3* | +0.05 - +0.5 | 0.001 |
| Lysine | 245.6 ± 11.7 | +38.1* | +0.9 - +75.4 | 300.3 ± 13.8 | +0.7 | -26.7 - +28.1 | 0.002 |
| Alanine | 221.8 ± 13.5 | +40.1* | +11.4 - +68.8 | 284.1 ± 18.3 | -40.7* | -72.8 - -8.7 | 0.003 |
| Glutamine | 537.3 ± 29.5 | -17.2 | -85.2 - +50.8 | 594.1 ± 26.7 | -89.4* | -140.7 - -38.1 | 0.007 |
| Threonine | 176.4 ± 14.8 | +30.7* | +0.9 - +60.4 | 177.8 ± 10.2 | -9.5 | -34.4 - +15.4 | 0.008 |
| Methionine | 25.0 ± 1.4 | -0.3 | -2.8 - +2.2 | 27.9 ± 1.0 | -3.0* | -4.9 - -1.3 | 0.01 |
| Fischer Ratio | 1.9 ± 0.8 | +0.1 | -0.01 - +0.4 | 1.7 ± 0.06 | +0.2* | +0.05 - +0.3 | 0.01 |
| Citrulline | 60.5 ± 5.2 | +11.5 | -6.8 - +29.7 | 89.4 ± 9.7 | -33.1* | -58.2 - -8.0 | 0.02 |
| Histidine | 20.8 ± 1.1 | +1.5 | -0.3 - +3.3 | 18.9 ± 1.1 | +3.0* | +0.7 - +5.3 | 0.02 |
| Tryptophan | 39.1 ± 3.1 | +5.8 | -0.6–12.2 | 48.3 ± 2.2 | +2.4 | -2.3 - +7.1 | 0.02 |
| Leucine | 65.4 ±3.8 | +9.5* | +3.7 - +15.4 | 71.7 ± 2.8 | +1.5 | -5.5 - +8.5 | 0.02 |
| Phenylalanine | 29.2 ± 1.9 | +3.2 | -2.6 - +9.0 | 35.8 ± 1.3 | -3.7 | -11.5 - +4.0 | 0.03 |

*denotes significant within group changes before and after intervention at $p \leqslant 0.002$.

Time x Group denotes significance of between group changes.

CI- confidence interval of within group changes, Fischer Ratio—ratio of sum of branched chain amino acids (valine, leucine, isoleucine) over sum of aromatic amino acids (phenylalanine, tyrosine). Arg Bioavail Raio–ratio of arginine to sum of citrulline and ornithine.

glucose (FBG; Fig 3C) were positively associated with percent changes in citrulline (Cit; r = 0.61, p = 0.003) and cystathionine (Cysth; r = 0.57, p = 0.009), and the arginine bioavailability index (ArgBioavail; r = 0.70, p < 0.001), assessed by ratio of arginine to sum of citrulline + ornithine. Cysth change also correlated with change in insulin (r = 0.57, p = 0.009), and accordingly with HOMA (r = 0.63, p = 0.003). A strong positive correlation emerged for ornithine alone with change in CRP (r = 0.63, p < 0.001). Consistent with the baseline analysis, changes in amino acids did not associate significantly with lipid parameters.

## Discussion

Improved nutrition represents a lifestyle modification target to lower the growing global cardiometabolic disease burden [19], especially among youth and their families struggling with socioeconomic disparity and the many associated barriers to a healthy diet [20]. Even without economic barriers, lifestyle adherence is challenging [21]. Furthermore, strict dietary counseling and adherence for the management of hyperlipidemia has been associated with increased anxiety and depression in youth [22]. The nutrient bar strategy utilized in this study was developed to enable research on the impact of nutrition on CMR, by facilitating daily delivery of nutrients from foods missing from a standard American diet, essential fibers, minerals, vitamins, omega-3 fatty acids, and polyphenols, in a readily consumed, low-calorie, palatable nutrient bar [8–10]. Results herein show that a 2 month nutrient bar intervention together with increased physical activity improved systolic blood pressure in Teens only without other significant changes observed in traditional biomarkers, but was associated with metabolomic

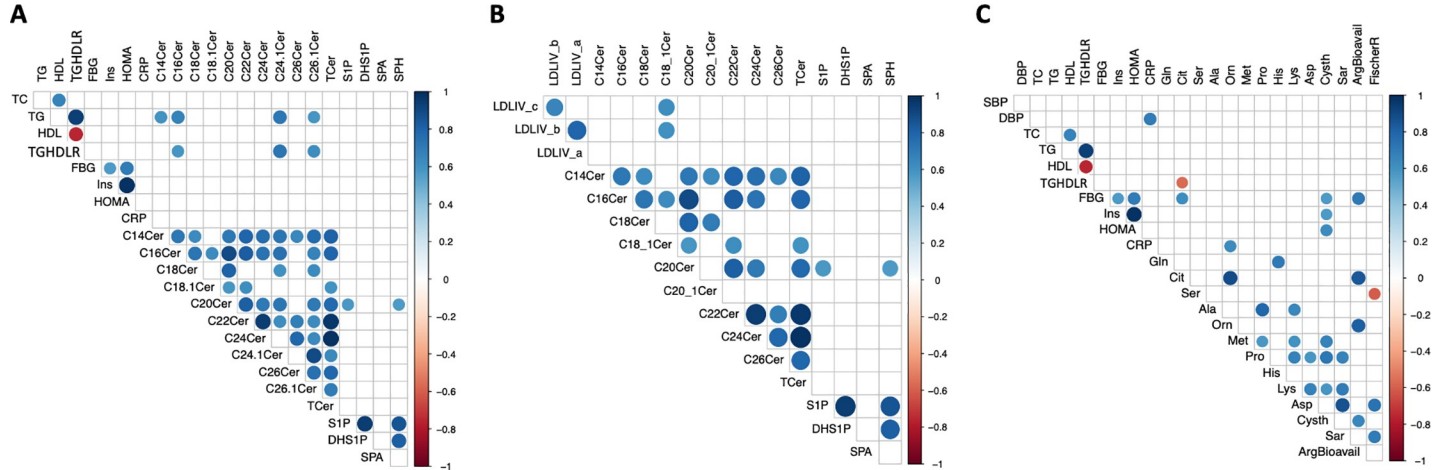

**Fig 3.** Correlograms for specific changes in ceramides with A. Traditional biomarkers of cardiometabolic risk (CMR) and with B. Lipoprotein subspecies. The correlogram for amino acid metabolites with traditional CMR biomarkers is shown in C. Bivariate correlations between percent changes in ceramide and amino acid metabolites with percent changes in lipid and insulin resistance parameters were performed using pearson correlation analysis. Correlations with *p* values less than 0.01 are shown. Right colored bar indicates scale range of pearson correlation coefficients depicted. Dark red denotes negative, whereas blue indicates positive correlations observed. Arginine bioavailability ratio (ArgBioavail) was calculated using molar ratios of arginine and sum of citrulline and ornithine. Fischer ratio (FischerR) was calculated using the ratios of molar sum of branched chain amino acids (valine, leucine, and isoleucine and aromatic amino acids (phenylalanine, and tyrosine). Abbreviations: SBP–systolic blood pressure, DBP–diastolic blood pressure, TC- total cholesterol; TG–triglyceride; LDL-low density lipoprotein cholesterol; HDL–high density lipoprotein cholesterol; TGHDLR–TG to HDL-cholesterol ratio, FGB–fasting blood glucose, Ins–insulin, HOMA–homeostatic model assessment of insulin resistance; CRP–high sensitive C-reactive protein; LDL_IVb and IV_a (very small LDL); Cer—Ceramide, T. Cer—Total Ceramide, SPM—Sphingomyelin, Gln—glutamine, Cit–citruilline, Ser–serine, Ala–alanine, Orn–ornithine, Met—Methionine, Pro–Proline, His–histidine, Cys–cysteine, Asp–aspartic acid, Cysth—cystathionine, Sar–sarcosine, DHSIP-Dihydrosphingosine-1-phosphate, SIP—Sphingosine-1-phosphate.

shifts in the full INT PAC and Teen group that correlate with favorable changes in cardiometabolic disease risk biomarkers.

Baseline metabolomic analysis identified ceramides as unique biomarkers of atherogenic dyslipidemia in both Teens and adults at heightened CMR (Fig 2). Although the metabolomic panel included BCAA and AAA previously implicated in CMR [23], only PC1, composed primarily of ceramide metabolites, associated with baseline TG levels and TG-rich VLDL subspecies, suggesting that ceramides may contribute to TG homeostasis. Fatty-acyl CoA is a substrate for both ceramide and TG biosynthesis and thus, excess ceramide and TG synthesis may occur under conditions of abundant fatty acid availability through both high fat diets or high refined carbohydrate diets and de novo lipogenesis.

In contrast to TGs that serve a passive fatty acid storage function, ceramides have unique signaling roles in mediating insulin resistance and mitochondrial bioenergetics [24]. Ceramides may therefore act as cellular energy sensors to regulate tissue fuel uptake and storage via insulin signaling. In obesity with chronic inflammation, excess ceramide, particularly C16:0 Cer, is associated with mitochondrial dysfunction, impaired sphingolipid cell signaling and insulin resistance [25]. Decreasing tissue ceramide through pharmacological or genetic manipulations can reverse or attenuate insulin resistance [26], providing support for a direct role of ceramides as critical part of fuel sensing circuitry in the body [27]. Ceramides may represent a sensitive biomarker of metabolic dysregulation when energetic supply chronically exceeds storage capacity. The current findings suggest that this tipping point may be influenced not only by absolute caloric intake but also by the dietary nutrient value.

Intensive lifestyle modification has been shown to improve CMR biomarkers in the obese [28]. In this trial, despite excellent participation in weekly exercise sessions, self-report of increased exercise between sessions in all subjects and both subjective and objective evidence

of compliance with nutrient bar intake in the INT group, there were minimal changes in traditional parameters of dyslipidemia, inflammation, and insulin resistance (Table 2). Changes in the plasma metabolome were however able to signify highly significant differences between CONT and INT (Tables 4, 5 and 6).

The nutrient supplement INT modified effects of physical activity on plasma ceramide metabolism. Contrary to the expectation that physical activity would lead to lower ceramide formation by improving lipid oxidation capacity, significant increases in C14:0, C16:0, C20:0 and C22:0 and overall increasing trends with all ceramide species were observed in CONT participants. A study by Bergman and co-workers similarly found that acute exercise transiently increases serum ceramides in obese untrained subjects [29]. Significant increases in sphinganine (Table 5) in CONT may reflect increasesd de novo synthesis driven in part by a higher rate of lipolysis following exercise. Exercise increases lipolysis [30] but given obesity related decreases in muscle fatty acid oxidation (FAO) [31], a rise in free palmitate following exercise may, in the short-term, favor increased ceramide synthesis. The observed rise in ceramides in CONT may therefore reflect transient increases in fatty acid mobilization. Further training which would result in an increased FAO capacity would be expected to normalize these parameters.

Effects of exercise on ceramides were largely blunted when subjects consumed daily nutrient bars, suggesting enhanced muscle FAO by the provision of nutrient substrates missing from a typical American diet. The nutrient bar is enriched in polyphenols and omega-3 fatty acids, known to increase muscle peroxisome proliferator activated receptor coactivator 1 alpha activity and mitochondrial biogenesis [32], processes which would be expected to improve amino acid and fatty acid catabolism. In support of this mechanism, we also observed significant lowering of non-essential amino acids (NEAA), serine, proline, aspartate, glutamine, and alanine (Table 6). Lower NEAA's may reflect improved substrate utilization and clearance of biomarkers associated with what has been termed the "metabolic gridlock" associated with obesity [33]. Furthermore, because serine is a substrate for synthesis of sphingolipids, its decrease may also have contributed to the lack of a rise in ceramide detected among INT participants. Alternatively, serine mobilization from other sites into plasma may be lessened when ceramide synthesis is not triggered.

Although clinical changes in TG, FBG, insulin and lipoproteins were not significant, we observed several significant correlations between changes in the plasma lipidome with changes in TG and physiologically interrelated pro-atherogenic small dense LDL particles (Fig 3). Plasma contains upwards of 200 distinct SPL species distributed across HDL, apoB-containing lipoproteins and albumin [34], but differential association with the most atherogenic lipoprotein subspecies has not been described. As with baseline correlations, the change to change correlations following the nutrient bar intervention were highly granular with specific ceramide species showing associations with changes in TG. Ceramides with C14:0, C16:0, C24:1 and C26:1 showed positive associations with changes in TG. C14:0 ceramide has been previously been implicated in nonalcoholic fatty liver disease (NAFLD) in adolescent children [35]. C16:0 ceramide inhibits mitochondrial FAO, oxidant production, and impairs Complex IV activity [25]. C24:1 ceramide significantly decreased in the INT group. Inhibition of synthesis of very long chain ceramides ($\geq$C22) by deletion of fatty acid elongase 6 protects against hepatic steatosis in high fat diet induced obesity [36]. Very small LDL particles showed a highly specific correlation with C18:1 ceramide (Fig 3B). Deletion of ceramide synthase 1 responsible for C18 ceramide synthesis has also been shown to increase mitochondrial FAO and protect against high fat diet induced NAFLD [37]. Interaction between specific ceramide species and small LDL may be involved in the progression from NAFLD to nonalcoholic steatohepatitis (NASH). NASH is not only characterized by high concentrations of small LDL and a low

mean LDL diameter but post hoc analyses from the Pioglitazone vs Vitamin E vs Placebo for the Treatment of Nondiabetic Patients with NASH (PIVENS) trial demonstrated that resolution of NASH was associated with an increase in the average LDL particle diameter and decreased small and very small LDL particles [38]. These lipoprotein subspecies parameters worsened in the participants who did not resolve their NASH. Larger LDL particles have improved affinity for LDL receptors and decreased retention in the subendothelial glycoprotein matrix, so would be expected to lead to decreased LDL and reduced atherogenesis. These findings collectively suggest that early changes in ceramide species among INT participants signal favorable shifts in lipid metabolism that may precede improvement in traditional clinical biomarkers.

In both CONT and INT participants, sphingosine-1-phosphate (S1P) levels increased, but with S1P levels rising higher among INT participants. A previous study showed that exercise increases plasma S1P levels and decreases erythrocyte ceramide levels and this effect was more evident in untrained subjects [39]. S1P is degraded by sphingosine-1-phosphate lyase (SPL). SPL dependent degradation of S1P is terminal step in sphingolipid degradation critical for regulating the pool size of all sphingolipids. In blood, ~50–70% of S1P is bound to apolipoprotein M within the HDL complex and is important for its reverse cholesterol transport [40]. Decreased S1P bound to HDL is associated with higher risk for coronary artery disease [41]. Although there were no absolute changes in HDL subspecies, quantification of HDL-bound S1P may provide further insights as to whether increases in plasma S1P in our study participants contribute to improved antiatherogenic HDL functions.

Correlation analysis with changes in amino acids showed highly specific associations between changes in citrulline, arginine bioavailability (ratio of arginine to sum of citrulline + ornithine) and cystathionine and changes in measures of insulin resistance. Obesity and insulin resistance are associated with increased CRP, increased citrulline and decreased arginine bioavailability in Teens [42]. This may reflect increased nitric oxide demand associated with the inflammatory metabolic stress of insulin resistance. Interestingly in this cohort with high baseline CRP, only change in ornithine correlated with change in CRP. Arginase expression and increased Ornithine/Arginine ratio has been shown to associate with vasculopathies in humans and in animals fed a high fat, high sugar Western diet [43]. The favorable pairwise difference in arginine bioavailability in the INT vs CONT groups may have contributed to subgroup specific improvement in systolic BP among INT Teens. Insulin sensitizing treatment with metformin plus pioglitazone for 3 months also improves arginine bioavailability in obese subjects with impaired glucose tolerance [44]. Change in Cystathionine (Cysth) was positively associated with change in glucose, insulin, and HOMA. Cysth is part of an alternate redox pathway to NOS, synthesized by cystathionine-beta-synthase (CBS) via the transsulfuration pathway. The expression of CBS in the liver observed in mice on a high fat diet has been proposed to be a defense mechanism against increased oxidative stress [45]. Decreased Cysth among INT participants may therefore reflect a lower burden of oxidative stress that could lead to improved insulin regulation.

Our study is limited by its relatively short observation period and the small sample size that precluded detailed analysis of data stratified by age. However, covariate analysis showed no significant effects of age with either the baseline correlations or with metabolic responses to the bar suggesting the metabolic impact of the nutrient bar intervention is conserved between the age groupings considered. The sample size was insufficient to consider hereditary factors. The findings in this predominantly minority study cohort may therefore not be generalizable. Furthermore, race and ethnicity, BMI and CRP differed by chance between INT and CONT groups although both groups exhibited comparable and considerable baseline CMR risk. The fact that lower total plasma Cers and specifically lower Cer 24:1 are described in persons of

African American ancestry [46] who by chance represented a larger proportion of the CONT group, cannot explain either the differential rise in total Cers among controls, nor the significant lowering of the Cer24:1 species in the INT group only. A larger study will be required to stratify analyses by race, ethnicity and other hereditary factors that may affect sphingolipid metabolism to determine if this may have influenced results.

In summary, our study has uncovered several granular and unique associations between plasma ceramides and amino acid metabolites with clinical parameters of dyslipidemia, inflammation and insulin resistance. These findings suggest that remediation of essential nutrients typically lacking in western style diets should be considered an essential component of preventive interventions directed towards the alleviation of CMR. In this regard, the metabolomic changes as monitored herein are more sensitive indicators of favorable, but subtle shifts in metabolism. A lack of response in traditional clinical CMR biomarkers does not necessarily reflect a lack of compliance with behavioral interventions. The positive associations between changes in established traditional risk factors and changes in metabolomic biomarkers support the hypothesis that metabolomic biomarkers are sensitive prognostic indicators at the leading edge of response to lifestyle therapy, possibly mediated at the level of mitochondrial function [47]. Further larger and longer studies are needed to validate whether the changes observed in these analyses will predict and precede future favorable changes in traditional CMR biomarkers and the benefits to weight regulation that can be anticipated with improved metabolism [48].

## Supporting information

**S1 Protocol. Institutional Review Board approved protocol.**
(PDF)

**S1 Flowchart. Flowchart detailing the study stages.**
(DOCX)

**S1 Fig. Scree plot that justifies retention of six principal components in the study analyses.**
(TIFF)

**S1 Data. Full de-identified data set of study measures.**
(XLS)

**S2 Data. Block Food Frequency dietary data.**
(XLS)

## Acknowledgments

The authors thank Rachelle Woods and the USDA ARS (Albany, CA) for assisting with nutrient bar production, the staff at the University of California Benioff Children's Hospital Oakland Pediatric Clinical Research Center and administrator Teresa Klask for invaluable assistance. We are indebted to Dr. Jichuan Wang, senior biostatistician in the Children's National Hospital Research Institute for his review of the full biostatistical analysis. The authors are also grateful for donations of Bar ingredients: Pharmachem Laboratories (Kearny, NJ, USA) for a vitamin and mineral blend; the U.S. Highbush Blueberry Council (Folsom, CA, USA) for blueberry powder; and Dow Chemical (Midland MI, USA) for hydroxypropylmethylcellulose.

## Author Contributions

**Conceptualization:** Michele Mietus-Snyder, Bruce N. Ames, Jung H. Suh.

**Data curation:** Michele Mietus-Snyder, Nisha Narayanan, Joyce C. McCann, Jung H. Suh.

**Formal analysis:** Ronald M. Krauss, Jung H. Suh.

**Funding acquisition:** Bruce N. Ames.

**Investigation:** Michele Mietus-Snyder, Nisha Narayanan, Kirsten Laine-Graves.

**Methodology:** Michele Mietus-Snyder, Jung H. Suh.

**Project administration:** Michele Mietus-Snyder.

**Resources:** Ronald M. Krauss, Jung H. Suh.

**Supervision:** Michele Mietus-Snyder.

**Validation:** Jung H. Suh.

**Visualization:** Jung H. Suh.

**Writing – original draft:** Michele Mietus-Snyder, Jung H. Suh.

**Writing – review & editing:** Michele Mietus-Snyder, Nisha Narayanan, Ronald M. Krauss, Kirsten Laine-Graves, Joyce C. McCann, Mark K. Shigenaga, Tara H. McHugh, Bruce N. Ames, Jung H. Suh.

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
