## [Decision Letter · Decision Letter 0]

17 Apr 2020

PONE-D-19-34911

Randomized Nutrient bar supplementation improves exercise-associated changes in plasma metabolome in adolescents and adult family members at cardiometabolic risk

PLOS ONE

Dear Prof. Michele Mietus-Snyder,

Thank you for submitting your manuscript to PLOS ONE. After careful consideration, we feel that it has merit but does not fully meet PLOS ONE’s publication criteria as it currently stands. Therefore, we invite you to submit a revised version of the manuscript that addresses the points raised during the review process.

We would appreciate receiving your revised manuscript by 30th July. To enhance the reproducibility of your results, we recommend that if applicable you deposit your laboratory protocols in protocols.io, where a protocol can be assigned its own identifier (DOI) such that it can be cited independently in the future. For instructions see: http://journals.plos.org/plosone/s/submission-guidelines#loc-laboratory-protocols

We look forward to receiving your revised manuscript.

Kind regards,

Massimiliano Ruscica, Ph.D.

Academic Editor

PLOS ONE

Reviewers' comments:

Reviewer's Responses to Questions

**Comments to the Author**

1. Is the manuscript technically sound, and do the data support the conclusions?

Reviewer #1: Yes

Reviewer #2: Yes

Reviewer #3: Partly

2. Has the statistical analysis been performed appropriately and rigorously? 

Reviewer #1: Yes

Reviewer #2: I Don't Know

Reviewer #3: No

3. Have the authors made all data underlying the findings in their manuscript fully available?

Reviewer #1: Yes

Reviewer #2: Yes

Reviewer #3: Yes

4. Is the manuscript presented in an intelligible fashion and written in standard English?

Reviewer #1: Yes

Reviewer #2: Yes

Reviewer #3: Yes

5. Review Comments to the Author

Reviewer #1: Dear Editor,

I carefully read the article by Mietus-Snyder et al., which is overall original and interesting. The manuscript is well-written and balanced in its parts and the abstract is informative.

Some comments for the Authors:

- The limitations of the study (e.g. the small sample size and the short observation period) should be addressed in the discussion

- In the discussion, Authors might refer to the article by Cicero et al. doi: 10.1016/j.ijcard.2019.05.069

- In Table 1, waist circumference values (and related parameters) should be reported separately for male and female

- Figure 4 should be improved because it is now unreadable

Reviewer #2: Major revision

The Authors compared cardiometabolic risk factors change after consumption of 2 nutrient bars daily for 2 months in overweight subjects to determine 1) cross-sectional relationships in both adolescents and adults between traditional CMR biomarkers and amino acid and ceramide metabolites and 2) longitudinal changes within groups in the same CMR biomarkers following the lifestyle +/- nutrient bar intervention.

- Study design: why the Authors chose a 2-month intervention when in the background they stated that, with the same bar, change in cardiometabolic biomarkers in obese adolescents can be seen only after 6 months?

- It is not completely clear which is the added value of the PAC group, even more so that they are not randomly assigned in CONT or INT group and that the inclusion criteria are different.

- how the group randomization occurred? CRP and BMI are very different, with INT group being less inflamed and less obese. The Author stated that, despite these differences, other parameters are similar thus suggesting close CMR profile. However, studies that you cited in the introduction support the efficacy differences according to CRP and BMI levels.

- Table 1. Why PAIRED t-test to compare Parent vs Teen in the same group? Unpaired t-test is more appropriate.

- In general, statistics notes in tables are not clear. It is not clear which groups the Authors are comparing. Line 126 is not clear, and it is not clear the meaning of each symbol used to denote statistically significant. b letter to denote significant differences is not defined in figure legend

- Please, use correct symbols for metric unit. Meter is not in capital letter, while liter and kilograms are.

- Line 223-224-225 the quality of self-report diets assessed by food frequency … however there is no data reported. It could be useful to add a questionnaire analysis (as supplemental)

- it is not clear why table 1 and table 2 are differently organized. In table 1 CONT vs INT are reported, while in table 2 TEEN and PAC.

Minor points:

- Please revise all the abbreviations along the paper. They should be defined the first time they appear (e.g. line 53 BMI). Please be consistent with abbreviations: Line 126 – what does “C” mean? Is it CONT group? Line 153 WHtR/WHRatio in table 1 and many others.

- Please revise typos (e.g. line 87 “adolecents”; line 89 “nutritient”)

Reviewer #3: The manuscript addresses an interesting topic. Several statistical methods are combined to achieve the goal and answer to the well-posed research questions. The results are also rather interesting, but several model's assumptions are overlooked. My overall feeling is that the data are of interest, but the work requires a deep statistical revision, possibly by an expert.

1. Rotation is more usually used in factor analysis than in PCA. PCA can be used as an ad hoc approximation to factor analysis. In factor analysis, the objective is to find a small number underlying common factors which generate the observed variables. It is assumed that each observed variable is generated as a linear function of the underlying factors, plus a unique random error term. The underlying factors are assumed to be orthonormal. The solution is a set of factor loadings (if there are m factors and n variables this will be an n by m matrix) and a set of n unique variances. The maximum likelihood solution is undefined down to an orthogonal transformation (i.e. rotations) of the factor loadings. That is, any rotatoion of the factor loadings is an equally good solutions.

So it makes sense to rotate these loadings in a way which maximises ease of interpretation. Different rotation algorithms are based on different approaches to what is easiest to interpret. If you rotate the solution of a PCA it is no longer a PCA! Please, clarify this point, as this is crucial to understand the procedure you are considering. According to the introduction of the GEE approach, I am more incline to believe that you considered a factor model. There are several methods for estimating the factor loadings and communalities, including the principal component method, principal factor method, the iterated principal factor method and maximum likelihood estimation. The principal component method is one of the most common approaches to estimation and will be employed on the rootstock data seen in previous posts. The principal component method is rather misleading in its naming that no principal components are calculated. Please, mention all the drawbacks in using oblique rotations. The results obtained by an oblique rotation will be less likely to be replicated by future studies (this occurs because of sampling error).

2. A graphical representation of the dimensionality reduction methods should be provided. Please, provide the scree plot and info on the proportion of variance explained by the components. Similarly, please provide the individual and variable graphs in the lower dimensional space. Maybe, I also miss something, and how the scores are computed is rather unclear. In principle, you have one score per-component; here, one common score is considered. Please, clarify.

3. The sample size is rather small. Please, provide evidence that the assumptions behind any performed statistical tests are fulfilled. This is also true for the GEE model. For the GEE approach, it is rather unclear which working matrix is considered and why. Both are crucial points. The authors must provide evidence that all the model's assumptions are fulfilled. Otherwise, inferential results are not reliable.

4. I cast some doubts on the computations provided in Table 1. The standard deviations (what does SEM mean?) for the "combined" data are rather unplausible, according the group-specific summary info.

6. PLOS authors have the option to publish the peer review history of their article (what does this mean?). If published, this will include your full peer review and any attached files.

Reviewer #1: No

Reviewer #2: No

Reviewer #3: No

---

## [Author Response · Author response to Decision Letter 0]

4 Aug 2020

A full letter with these comments has been uploaded but they are copied again here.

TITLE: Randomized Nutrient bar supplementation improves exercise-associated changes in plasma metabolome in adolescents and adult family members at cardiometabolic risk 

Dear Dr. Ruscica,

Thank you for the opportunity to respond to the constructive reviews provided to our submission to PLOS One. We appreciate your generous window through 7/30/20. We are grateful that reviewers 1 and 2 agree that this is a technically sound piece of research and hope we have satisfied reviewer 3’s concerns about the rigor of the statistical assessment. We appreciate that all three reviewers agree we have been transparent with provision of the data on which our conclusions are based and have presented the results in intelligible language. We have performed all of the statistical analyses again with oversight from a senior biostatistician and have addressed all reviewer comments in the revised manuscript now respectfully submitted as both marked (with tracked edits) and unmarked (edits accepted) versions. We have made every effort to comply with the PLOS One Style requirements in accord with journal templates.

Before addressing the specific reviewer comments point-by-point below, we would like to respond to your editorial request to avoid reporting any “data not shown”. We have removed this phrase from the manuscript in the single place it had been inappropriately used (line 284). The preceding statement stated the lack of significant correlation for specific relationships between variables listed, and are of course supported by the full data set already provided as a supplemental table. We are adding to the original supplemental data set, an additional data set with our Block FFQ nutrition data, and the PC analysis scree plot, per reviewer request. 

Our full responses are embedded below in bold to each of the reviewer comments and referenced by page/line of the resubmission.

5. Review Comments to the Author

REVIEWER #1: Dear Editor,

I carefully read the article by Mietus-Snyder et al., which is overall original and interesting. The manuscript is well-written and balanced in its parts and the abstract is informative.

Some comments for the Authors:

- The limitations of the study (e.g. the small sample size and the short observation period) should be addressed in the discussion 

Response: We thank the reviewer for the comment on the overall original and interesting nature of this work and agree that our small sample size and relatively short observation period pose significant limitations. We had recognized these things in the discussion of the paper, opening the paragraph on study limitations with the statement (p 26 line 22) “Our study is limited by its small sample size…” and we have added the limitation also posed by the short observation period. We had already concluded that a prospective study with a longer observation period will be necessary to validate the hypotheses generated by this work but have added in accord with the reviewer’s accurate critique that future validation studies will also need to be larger (p27, lines 22). 

- In the discussion, Authors might refer to the article by Cicero et al. doi: 10.1016/j.ijcard.2019.05.069 

Response: Thank you for making us aware of this interesting article. Although it refers to a familial hypercholesterolemia cohort rather than the mixed dyslipidemia under consideration in our study, it adds important evidence to the challenge of motivating healthy dietary behavioral change in children. We have added this reference (#22) at the opening to our Discussion (p 22, lines 1-3).

- In Table 1, waist circumference values (and related parameters) should be reported separately for male and female 

Response: The reviewer’s point is well taken given average waist circumference based upon population data is greater in males as compared with females, but only one of the 18 Parent Adult Caretakers (PAC) was male. His WC of 98.0 cm actually fell below the adult female PAC average WC (108.8 + 15.6 cm) but well within the full range observed (81.0 to 136.7 cm). We did compare this metric for male (106.3 + 19.2 cm) vs. female (103.8 + 11.8 cm) TEENS and no significant difference was found (p = 0.72). Because the effort to present baseline data for both adult PAC and Teens in both INT and CONT groups already necessitates extensive baseline and follow-up tables 1 & 2, we respectfully did not add this additional categorization. We do however now mention our rationale – namely that WC did not differ by sex in this study cohort (p 12, lines 4-8)- and include that BMI did not differ by sex either (p 11, line 22). 

- Figure 4 should be improved because it is now unreadable 

Response: Figure 4 – now numbered as Figure 3 - has been regenerated with the new PC analyses at improved resolution. We have also reorganized Table 4 into Table 4a (for differential ceramide changes between Control and Intervention groups) and 4b (for changes in sphingolipid bases).

REVIEWER #2: Major revision

The Authors compared cardiometabolic risk factors change after consumption of 2 nutrient bars daily for 2 months in overweight subjects to determine 1) cross-sectional relationships in both adolescents and adults between traditional CMR biomarkers and amino acid and ceramide metabolites and 2) longitudinal changes within groups in the same CMR biomarkers following the lifestyle +/- nutrient bar intervention.

- Study design: why the Authors chose a 2-month intervention when in the background they stated that, with the same bar, change in cardiometabolic biomarkers in obese adolescents can be seen only after 6 months? 

Response: This was the first trial of the nutrition bar in adolescents, temporally preceding the study of obesity asthma in adolescents referenced by the reviewer. We did not therefore know that traditional biomarkers would not respond within 2 months. Indeed, it was because of our lack of movement in traditional biomarkers at 2 months, that the subsequent trial in adolescent asthma was extended for a full 6 months, while we began exploring the post hoc analyses for change in sphingolipid ceramide biomarkers that may precede traditional biomarkers, as reported upon in this work. Unfortunately, neither metabolomics or lipidomics were performed on the longer adolescent trial. 

- It is not completely clear which is the added value of the PAC group, even more so that they are not randomly assigned in CONT or INT group and that the inclusion criteria are different. 

Response: Their inclusion was primarily to reinforce family level change for the adolescents – and we believe this study design contributed to the unusually good adherence and compliance observed. 

- how the group randomization occurred? CRP and BMI are very different, with INT group being less inflamed and less obese. The Author stated that, despite these differences, other parameters are similar thus suggesting close CMR profile. However, studies that you cited in the introduction support the efficacy differences according to CRP and BMI levels. 

Response: Although BMI and level of CRP certainly correlate, the differences in response to the nutrition bar previously described in adults were associated with the level of inflammation rather than BMI. It is important to note that any level of CRP elevation over 1.5 mg/L (SI units) was sufficient to blunt the early response to the nutrition bar without an obvious dose relationship. While the CRP, a correlate as noted of BMI, was significantly higher in the CONT vs INT cohorts in this study, both groups had a mean CRP exceeding 1.5 mg/L (= 14.3 nmol/L). As noted above, this was the first trial of this nutrient intervention in adolescents and it was not clear, a priori, whether or not younger age would negate the previously observed effect of inflammation on response. None-the-less we recognize and have noted in limitations that the imperfect randomization scheme in this small cohort. The lower end of the range of BMIs was equivalent in both cohorts (INT 25.2 to 46.5; CONT 25.1 to 50.7) but one dyad randomized to the CONT group with higher BMI’s of 48.9 and 50.7 for the adolescent and adult respectively, did skew the higher range limit and therefore the mean. If this one dyad is not included, the range of BMIs for the CONT group becomes 25.1 to 43.4 and there is no longer a significant difference between the two groups in BMI. It is a significant limitation as noted (p 27, lines 4-6), but we hope the reviewers agree, the findings remain provocative and worthy of further study.

- Table 1. Why PAIRED t-test to compare Parent vs Teen in the same group? Unpaired t-test is more appropriate. 

Response: The Statistical Methods had been written with an emphasis on the pre-post analyses of changes in metabolites that were compared by paired t-tests. We have added clarification that the group comparisons in all initial descriptive analyses and any subsequent between group comparisons were done, as the reviewer correctly notes they should be done, with unpaired t-tests. 

- In general, statistics notes in tables are not clear. It is not clear which groups the Authors are comparing. Line 126 is not clear, and it is not clear the meaning of each symbol used to denote statistically significant. b letter to denote significant differences is not defined in figure legend 

Response: Thank you for a careful review of the tables and for identifying this lack of clarity. We have focused the statistical notes in the tables in efforts to describe significant differences more clearly. We no longer us letters a & b but simply one or two Asterix. The legend for Table 1 currently reads:

* Denotes statistically significant difference (p < 0.05) by unpaired student’s t-test between PAC (Parent Adult Caretakers) and Teen subgroups within Control (CONT) and/or Intervention (INT) groups.

** Denotes statistically significant difference (p <0.05) for the designated variable by unpaired student’s t-test between combined CONT and INT groups 

And the legend for Table 2 currently reads:

* Denotes statistically significant difference (p < 0.05) by paired student’s t-test within a group.

** Denotes statistically significant (p < 0.05) time by group interactions for the designated variable for both PAC and Teens (Vitamin D levels) unless specified (T for Teens only, SBP)

- Please, use correct symbols for metric unit. Meter is not in capital letter, while liter and kilograms are. 

Response: Thank you for this correction. These metric unit symbols have been corrected in both Tables and text.

- Line 223-224-225 the quality of self-report diets assessed by food frequency … however there is no data reported. It could be useful to add a questionnaire analysis (as supplemental) 

Response: We have added additional descriptive data from the self-report diets to document the small number of significant changes and to better document the absence of substantive change in dietary behavior (p 12, lines 12-17). We also include the dietary data derived from the Block Food Frequency Questionnaires now as well as an additional supplementary data table of these analyses.

- it is not clear why table 1 and table 2 are differently organized. In table 1 CONT vs INT are reported, while in table 2 TEEN and PAC. 

Response: We thank the reviewer for noting this unnecessary change in formatting a complex set of data. We have now organized both Table 1 and Table 2 in the same order.

Minor points:

- Please revise all the abbreviations along the paper. They should be defined the first time they appear (e.g. line 53 BMI). Please be consistent with abbreviations: Line 126 – what does “C” mean? Is it CONT group? Line 153 WHtR/WHRatio in table 1 and many others.

Response: We have carefully reviewed the full text and to ensure that all abbreviations are defined at first appearance and that they are used consistently in the text tables, and figures.

- Please revise typos (e.g. line 87 “adolecents”; line 89 “nutritient”)

Response: Both typos are corrected. 

REVIEWER #3: The manuscript addresses an interesting topic. Several statistical methods are combined to achieve the goal and answer to the well-posed research questions. The results are also rather interesting, but several model's assumptions are overlooked. My overall feeling is that the data are of interest, but the work requires a deep statistical revision, possibly by an expert.

Response: The senior author, Dr. Jung Suh, who has an MPH in biostatistics, performed the original statistical analyses but in light of this reviewer concern, Dr. Suh repeated the full metabolomic and lipidomic analysis with consultation from a senior biostatistician in the first author’s institution, Dr. Jichuan Wang, whose expertise includes structural equation modeling, multilevel modeling, and principal components analyses for large data sets. His valuable contribution to this resubmission is now acknowledged, and the input reflected in the comments below.

1. Rotation is more usually used in factor analysis than in PCA. PCA can be used as an ad hoc approximation to factor analysis. In factor analysis, the objective is to find a small number underlying common factors which generate the observed variables. It is assumed that each observed variable is generated as a linear function of the underlying factors, plus a unique random error term. The underlying factors are assumed to be orthonormal. The solution is a set of factor loadings (if there are m factors and n variables this will be an n by m matrix) and a set of n unique variances. The maximum likelihood solution is undefined down to an orthogonal transformation (i.e. rotations) of the factor loadings. That is, any rotatoion of the factor loadings is an equally good solutions.

So it makes sense to rotate these loadings in a way which maximises ease of interpretation. Different rotation algorithms are based on different approaches to what is easiest to interpret. If you rotate the solution of a PCA it is no longer a PCA! Please, clarify this point, as this is crucial to understand the procedure you are considering. 

According to the introduction of the GEE approach, I am more incline to believe that you considered a factor model. There are several methods for estimating the factor loadings and communalities, including the principal component method, principal factor method, the iterated principal factor method and maximum likelihood estimation. The principal component method is one of the most common approaches to estimation and will be employed on the rootstock data seen in previous posts. The principal component method is rather misleading in its naming that no principal components are calculated. Please, mention all the drawbacks in using oblique rotations. The results obtained by an oblique rotation will be less likely to be replicated by future studies (this occurs because of sampling error).

Response: In the revised manuscript, the PCA was implemented in the following way: First, a PCA model with no rotation was conducted to just extract the components. Second, the meaningful components were identified by checking eigenvalues and scree plot. Third, a PCA model with oblique rotation was tested to examine the factor correlation matrix. Since none of the factor correlations were over ± 0.32, indicating factor correlations are not driven by the data, we finally estimated a model with orthogonal rotation (that assumes the factors are not correlated) to reveal simple structure so that the factor loadings are easy to interpret.

2. A graphical representation of the dimensionality reduction methods should be provided. Please, provide the scree plot and info on the proportion of variance explained by the components. Similarly, please provide the individual and variable graphs in the lower dimensional space. Maybe, I also miss something, and how the scores are computed is rather unclear. In principle, you have one score per-component; here, one common score is considered. Please, clarify.

Response: In response to the reviewer’s comment, the scree plot is included with the supplemental materials of the revised manuscript. Because the elbow of the curve on this scree plot is closer to 6 rather than 5 principle components, we retained 6 components for further analysis. The Scree plot is included in the uploaded Response to Reviewers and is included with Supporting Data as part of this resubmission. It would not attach in this window.

The new PCA analysis changes Table 3 slightly (the full listing of the principal components), and therefore all analyses involving these components were redone, as were Figures 2 and 4 (the latter now becomes Figure 3 as noted below). All of the main findings of the original submission are confirmed. Table 3, due to increased width with a 6th PC included, is added as a screen shot to the manuscript with tracked edits. It has also been added for the Editors to the resubmission as a Word file under category of “other”. 

3. The sample size is rather small. Please, provide evidence that the assumptions behind any performed statistical tests are fulfilled. This is also true for the GEE model. For the GEE approach, it is rather unclear which working matrix is considered and why. Both are crucial points. The authors must provide evidence that all the model's assumptions are fulfilled. Otherwise, inferential results are not reliable.

Response: Yes, the sample size is small. However, model estimation terminated normally. We had already explained in the manuscript that this was a post-hoc analysis undertaken when the anticipated change in traditional biomarkers was not observed. The findings described stand up to rigorous analysis. As noted above, we had acknowledged in the original manuscript the importance of validating the reported observations in a larger prospective study cohort. The small sample size is mentioned as a limitation of the study in the Discussion section.

The reviewer points out an important issue, i.e., working matrix specified for GEE model estimation. Various residual correlation structures (e.g., autoregressive, exchangeable, and unstructured) are available for GEE, and the model provides robust parameter estimates even though the correlation structure may be incorrectly specified. As a common practice, we specified an unstructured correlation structure in our GEE model.

We agree with the reviewer that oblique rotations have some drawbacks. As the reviewer points out, the results obtained by an oblique rotation will be less likely to be replicated by future studies. In addition, the definition for a factor loading is different depending on whether it is in a factor pattern matrix or in a factor structure matrix. Because of correlated factors, the factor loadings are not equivalent to the correlations between the observed variables and the factors. In the revised manuscript, orthogonal rather than oblique rotation was implemented (see responses to Comment 1).

4. I cast some doubts on the computations provided in Table 1. The standard deviations (what does SEM mean?) for the "combined" data are rather unplausible, according the group-specific summary info.

Response: The abbreviation SEM refers to Standard Error of the Mean and should have been defined. We agree with the reviewer that Standard Deviation (SD) provides a better indication of dispersion of data around the mean for normally distributed continuous variables, and now report the SD for all outcomes reported in Tables 1 and 2, as well as the absolute metabolite data reported in Table 4a. 

We thank again all reviewers for their helpful critiques that have strengthened this research report.

---

## [Decision Letter · Decision Letter 1]

15 Sep 2020

PONE-D-19-34911R1

Randomized Nutrient bar supplementation improves exercise-associated changes in plasma metabolome in adolescents and adult family members at cardiometabolic risk

PLOS ONE

Dear Dr. Mietus-Snyder,

Thank you for submitting your manuscript to PLOS ONE. After careful consideration, we feel that it has merit but does not fully meet PLOS ONE’s publication criteria as it currently stands. Therefore, we invite you to submit a revised version of the manuscript that addresses the points raised during the review process.

We look forward to receiving your revised manuscript.

Kind regards,

Massimiliano Ruscica, Ph.D.

Academic Editor

PLOS ONE

Reviewers' comments:

Reviewer's Responses to Questions

**Comments to the Author**

1. If the authors have adequately addressed your comments raised in a previous round of review and you feel that this manuscript is now acceptable for publication, you may indicate that here to bypass the “Comments to the Author” section, enter your conflict of interest statement in the “Confidential to Editor” section, and submit your "Accept" recommendation.

Reviewer #1: All comments have been addressed

Reviewer #3: (No Response)

2. Is the manuscript technically sound, and do the data support the conclusions?

Reviewer #1: Yes

Reviewer #3: Yes

3. Has the statistical analysis been performed appropriately and rigorously? 

Reviewer #1: Yes

Reviewer #3: Yes

4. Have the authors made all data underlying the findings in their manuscript fully available?

Reviewer #1: No

Reviewer #3: Yes

5. Is the manuscript presented in an intelligible fashion and written in standard English?

Reviewer #1: Yes

Reviewer #3: Yes

6. Review Comments to the Author

Reviewer #1: Authors improved their manuscript following my suggestions. I have no more comment on it and suggest the warmly acceptation of the paper for publication.

Reviewer #3: I really appreciate the efforts to answer to my points. Detailed answered were provided, showing a deep knowledge on the statistical methods employed.

Only one minor, but substantial, point is still overlooked. The authors ackwonledge that the small sample size is a limitation of the study. However, all the employed tests are based on some assumptions (like the normality of the data, but not only) which must be checked to ensure a reliable inference. In the main text, I read "Continuous physical and metabolic variables were tested for normality by examining the skewness, kurtosis and the Shapiro Wilk tests and transformed as necessary before analysis". Please, provide evidence of the normality of the data and make clear which transformations are adopted and why.

7. PLOS authors have the option to publish the peer review history of their article (what does this mean?). If published, this will include your full peer review and any attached files.

Reviewer #1: No

Reviewer #3: No

---

## [Author Response · Author response to Decision Letter 1]

24 Sep 2020

RESPONSE: In real-life datasets, variables do not always follow the normal distribution. All continuous physical and metabolic variables in our study were tested for normality. The results of the Shapiro-Wilk tests show that most of the variables in our data were normally distributed, but log transformations were conducted for the 11 of 31 continuous physical and metabolic variables that were skewed to make them as normal as possible. The results of the Shapiro-Wilk tests show (with p-value > 0.05) that all of the transformed variables except two (i.e. small and medium LDL particle concentrations) are approximately normal. 

The measures of metabolite concentrations used for principal component analysis (PCA) were Z-transformed to render them normally distributed on the same scale with mean of zero and standard deviation of one. In addition, the generalized estimating equation (GEE) model used for our longitudinal data analysis is robust to data non-normality. The normality assumption of linear regression is about residuals [ε~ i.i.d. N(0, σ²)], and the GEE model does not necessarily assume normality of residuals.

We hope that this explanation will satisfy reviewer 3’s last remaining concern. We have added the following wording (p10, lines 12-18) to the current statement (in quotes here) to clarify this important point: 

 "Continuous physical and metabolic variables were tested for normality by examining the skewness, kurtosis and the Shapiro Wilk tests and transformed as necessary before analysis." Most of the variables in our data were normally distributed. Log transformations were conducted for the continuous physical and metabolic variables that were skewed to make them as normal as possible. The Shapiro-Wilk tests show (with p-value > 0.05) that all of the transformed variables except two (i.e. small and medium LDL particle concentrations) are approximately normal. The measures of metabolite concentrations used for principal component analysis (PCA) were Z-transformed to render them normally distributed on the same scale with mean of zero and standard deviation of one. 

We thank you for your prompt consideration. This manuscript will support a pending grant review to further probe these findings if available for the study section to view when they meet mid-October.

---

## [Decision Letter · Decision Letter 2]

28 Sep 2020

Randomized Nutrient bar supplementation improves exercise-associated changes in plasma metabolome in adolescents and adult family members at cardiometabolic risk

PONE-D-19-34911R2

Dear Dr. Michele Mietus-Snyder,

We’re pleased to inform you that your manuscript has been judged scientifically suitable for publication and will be formally accepted for publication once it meets all outstanding technical requirements.

Kind regards,

Prof. Massimiliano Ruscica

Academic Editor

PLOS ONE

Additional Editor Comments (optional):

Reviewers' comments:

Reviewer's Responses to Questions

**Comments to the Author**

1. If the authors have adequately addressed your comments raised in a previous round of review and you feel that this manuscript is now acceptable for publication, you may indicate that here to bypass the “Comments to the Author” section, enter your conflict of interest statement in the “Confidential to Editor” section, and submit your "Accept" recommendation.

Reviewer #3: All comments have been addressed

2. Is the manuscript technically sound, and do the data support the conclusions?

Reviewer #3: (No Response)

3. Has the statistical analysis been performed appropriately and rigorously? 

Reviewer #3: (No Response)

4. Have the authors made all data underlying the findings in their manuscript fully available?

Reviewer #3: (No Response)

5. Is the manuscript presented in an intelligible fashion and written in standard English?

Reviewer #3: (No Response)

6. Review Comments to the Author

Reviewer #3: (No Response)

7. PLOS authors have the option to publish the peer review history of their article (what does this mean?). If published, this will include your full peer review and any attached files.

Reviewer #3: No

---

## [Editor Report · Acceptance letter]

30 Sep 2020

PONE-D-19-34911R2 

Randomized Nutrient bar supplementation improves exercise-associated changes in plasma metabolome in adolescents and adult family members at cardiometabolic risk 

Dear Dr. Mietus-Snyder:

I'm pleased to inform you that your manuscript has been deemed suitable for publication in PLOS ONE. Congratulations! Your manuscript is now with our production department. 

Kind regards, 

on behalf of

Dr. Massimiliano Ruscica 

Academic Editor

PLOS ONE